# Rethinking procedural pain in labor: A comparison of lidocaine injection techniques for epidural catheter placement assessed with an objective clinician-centric pain score—A double-blind randomized controlled trial

Lukas Croner[1☉], Plato Lysandrou[2☉], Haosheng Li[1☉], Andrew Devine[3☉], Tyler Balon[1‡], Yun Xia[2‡], Nasir Hussain[2‡], Yue Yu[4☉], Mahmoud Abdel-Rasoul[4‡], Marco Echeverria Villalobos[2‡], Alberto Uribe[2‡], Elvia Vera Miquilena[2‡], Ling-Qun Hu[2☉*]

1 College of Medicine, The Ohio State University, Columbus, Ohio, United States of America,
2 Department of Anesthesiology, The Ohio State University College of Medicine, Columbus, Ohio, United States of America, 3 Heritage College of Osteopathic Medicine, Ohio University, Athens, Ohio, United States of America, 4 Center for Biostatistics, College of Medicine, The Ohio State University, Columbus, Ohio, United States of America

☉ These authors contributed equally to this work.
‡ TB, YX, NH, MA-R, MEV, AU and EVM also contributed equally to this work.
* lingqun.hu@osumc.edu

## Abstract

### Introduction

Local lidocaine infiltration before Tuohy needle insertion is essential for epidural analgesia. Lidocaine can be administered intradermally or subcutaneously, but the technique that causes less pain for laboring patients is unclear. Pain is typically assessed using the subjective Numeric Rating Scale (NRS), while the Critical-Care Pain Observation Tool (CPOT) offers an objective alternative, evaluating facial expressions, body movements, muscle tension, and vocalizations. This pilot study compared subcutaneous (SC) and intradermal (ID) lidocaine administration to evaluate lidocaine injection pain and its analgesic efficacy.

### Methods

In this double-blind randomized trial, laboring parturients received 3 mL of 1% lidocaine via SC (90-degree angle) or ID (60-degree angle) injection using a one-inch 25G needle. Primary outcomes included procedural pain during lidocaine administration, assessed using CPOT (clinician-centric) scores. Secondary outcomes encompassed lidocaine's analgesic efficacy during Tuohy needle insertion with both CPOT and NRS, hemodynamic stability, patient satisfaction, and NRS for two lidocaine injection techniques as references.

**Data availability statement:** All relevant data are within the paper and its Supporting information files.

**Funding:** Lukas Croner and Tyler Balon received the Medical Student Research Scholarship from The Ohio State University College of Medicine. Haosheng Li and Andrew Devine received funding from the Foundation for Anesthesia Education and Research (FAER) - Medical Student Anesthesia Research Fellowships (MSARF).

**Competing interests:** Neither funding source had any influence on the study design, methodology, data analysis, results, or conclusions. The other authors did not receive direct funding for this research.

## Results

Fifty-one patients were randomized into the SC Group (n = 26) and the ID Group (n = 25). No significant differences were observed in overall CPOT or NRS scores between groups, but SC administration yielded significantly lower muscle tension scores (Krushkal-Wallis test p = 0.018). The analgesic efficacy on Tuohy needle insertion, patient satisfaction, and hemodynamic values was not significantly different between the two techniques. A weak correlation between CPOT and NRS scores (Spearman's r = 0.32, p = 0.024) highlighted the complementary roles of objective and subjective, patient-centric, pain assessments. There was no statistical significance of interobserver variation for CPOT assessment.

## Conclusion

This pilot trial establishes proof of concept for validating the CPOT in obstetric settings and highlights both the need for and feasibility of future studies aimed at optimizing lidocaine administration protocols during labor epidural placement. While this study found no global differences in pain scores between subcutaneous and intradermal lidocaine, subcutaneous injections demonstrated less muscle tension with similar analgesic efficacy. The discordance between CPOT and NRS underscores the value of integrating both tools for comprehensive procedural pain evaluation.

## Introduction

Needle phobia affects an estimated 4% of people, creating healthcare barriers and avoidable obstetric complications. Perceived injection pain and the prospect of multiple needle insertions often deter patients with this phobia from critical interventions [1–4]. Research suggests local subcutaneous anesthetic injections may cause less pain than intradermal injections while providing comparable analgesia by targeting deeper dermal nerve roots [5]. Although methods like buffering lidocaine with sodium bicarbonate, warming the solution, using microneedles, surface applications of local anesthetics onto the skin before injection, and waiting time can reduce the pain of intradermal injections, no direct comparison between subcutaneous and intradermal approaches exists for laboring women [6–13]. Despite some evidence favoring subcutaneous injection, intradermal injection remains more commonly used despite potentially greater discomfort.

The lidocaine injection preceding Tuohy needle insertion is often the most painful step, triggering involuntary movements or nocebo-induced hyperalgesia, which can disrupt procedural accuracy and patient-provider focus. Pain assessment benefits from combining subjective NRS scales with the Critical-Care Pain Observation Tool (CPOT), which objectively quantifies behavioral responses like facial expression, muscle tension, body movements, and vocalizations (threshold: 2/8). CPOT can help address limitations of self-reporting, particularly in laboring patients who may be undercommunicating pain. CPOT, validated in conscious individuals, enhances

reliability by capturing nonverbal cues and correlating moderately with subjective scales, offering a holistic view of clinically significant pain, especially for care providers during interventions [14–19]. Since CPOT has not been validated in parturients or for epidural catheter placements, there is no existing data for our sample size estimation.

Due to the novel use of CPOT in parturient patients, we conducted this pilot study to compare the pain experienced during intradermal versus subcutaneous lidocaine injections using both the NRS and CPOT assessment tools, along with hemodynamic responses. This information will help facilitate future investigations. Additionally, the study aims to evaluate the analgesic efficacy of lidocaine on Tuohy needle insertion and overall patient satisfaction among laboring women requesting epidural analgesia. Although this is a pilot study, we preliminarily hypothesized that parturients receiving subcutaneous local lidocaine would experience lower pain scores than those receiving intradermal injection. This hypothesis was explored to inform the design of a future adequately powered study.

## Methods

This study is a prospective, single-center, double-blind, randomized pilot clinical trial conducted at our institution. Participants were randomized into two groups: the Intradermal (control, ID Group) and the Subcutaneous (study, SC Group) groups. Patients and the public were not involved in the design, conduct, reporting, or dissemination plans of this research. The study was approved by the local Institutional Research Board at The Ohio State University (Office of Responsible Research Practices, Protocol ID: 2023H0250) on Aug 28, 2023, and its registration was initiated at ClinicalTrials.gov (NCT06236126) on November 13, 2023. The study officially started on February 1st, 2024. Written informed consent was obtained from all participants. Inclusion criteria were patients ≥18 years old, parturients requesting neuraxial labor analgesia (epidural, combined spinal-epidural, or dural patch epidural), and the ability to provide informed consent in English. Exclusion criteria included administration of opioids within four hours before epidural catheter placement, administration of IV magnesium sulfate within the last 24 hours, diabetes mellitus (type I and II), neurocardiogenic signs or symptoms (e.g., dizziness, light-headeness, bradycardia, or syncope) during IV cannulation, cervical dilation > six cm, spinal analgesia alone, chronic pain patients, opioid use disorder, intrauterine fetal demise, and prisoners. The primary outcome was CPOT during lidocaine injections. Secondary outcomes included NRS score for lidocaine injection, the analgesic efficacy of the lidocaine injection on Tuohy needle insertion, patient satisfaction following the epidural catheter placement, and hemodynamic changes (heart rate and blood pressure) measured at baseline, after the lidocaine injection, and after Tuohy needle insertion.

We estimated that 25 patients per group (ID Group and SC Group) would be required to learn more about procedural pain in the laboring women population in this pilot study. We planned to consent up to 60 subjects in total, accounting for 20% of screen failure and/or early termination. After obtaining informed consent, participants were randomized on a 1:1 ratio into two groups: **ID Group** or **SC Group** for lidocaine administration using the automated Research Electronic Data Capture (REDCap) system. The blinding plan process was maintained by assigning unblinded and blinded roles to research personnel involved in the study procedures. Research participants and study personnel involved in CPOT, NRS, patient satisfaction assessments, and vital signs were blinded throughout the study. To avoid accidental unblinding events, blinded observers were stationed in front of the patient and collected the required data. Only anesthesia care providers performing the lidocaine injection were unblinded because they were the ones administering the randomized lidocaine injection technique.

Right before the insertion of the Tuohy needle, each patient received 3 mL of 1% lidocaine administered by an unblinded American Board of Anesthesiology board-certified anesthesiologist using a one-inch 25-G needle over 3 seconds between uterine contractions. In the ID Group, lidocaine was administered with the needle positioned at a 60-degree angle to the skin, with successful intradermal administration confirmed by the formation of a visible wheal or bleb beneath the skin, followed by deeper infiltration. In the SC Group, lidocaine was administered with the needle positioned at a 90-degree angle, with an initial deep infiltration followed by subcutaneous and then intradermal injection of 0.5–1.0 mL of

lidocaine, with or without appearance of a skin wheel. The anesthesiologist provider was the only person allowed to say "Here comes the lidocaine injection" before the injection to help prevent nocebo effects.

Once participants were positioned to receive the lidocaine injection, a blinded observer was stationed in front of the patient to collect data during a period without uterine contractions to avoid accidental unblinding events. During the procedure, the observer assessed and recorded the patient's vocal responses, facial expressions, bodily movements, and muscle tension (reflecting pain reflexes) for the CPOT, NRS scores, HR, and BP were assessed immediately following both the lidocaine injection and the Tuohy needle insertion. After the procedural anesthesiologist exited the room, the blinded observer collected a patient satisfaction score on a scale of zero (worst) to ten (highest). Following these measurements, the study procedures were considered completed.

## Statistical analysis

Patient demographics and clinical characteristics were summarized using descriptive statistics. Continuous variables were expressed as means with standard deviations or medians with interquartile ranges (IQR), while categorical variables were reported as frequencies and proportions. Demographics and clinical characteristics were compared between the study groups using a two-sample t-test or Kruskal-Wallis test for continuous variables and Chi-square test or Fisher's exact test for categorical variables. For primary aims, Kruskal-Wallis tests were used to test if there was a difference in CPOT score following lidocaine injection between the two study groups. For secondary aims, Kruskal-Wallis tests were used to assess the difference between the study groups in CPOT and NRS scores following Tuohy needle insertion, NRS score following lidocaine injection, as well as patient satisfaction scores. For hemodynamics, linear mixed-effects models were used to account for measurements at three time points per participant: baseline, after lidocaine injection, and after Tuohy needle insertion. Inter-examiner bias and inter-provider bias were examined using Kruskal-Wallis tests. All statistical tests were 2-sided, and a value of P less than 0.05 was used to indicate statistical significance. All statistical analyses and plots were performed using SAS 9.4 (SAS Institute, Inc.; Cary, North Carolina, USA).

## Results

Between November 21, 2023, and August 26, 2024, we enrolled 58 laboring women at the Ohio State University Wexner Medical Center, with seven patients not being included due to failure to meet inclusion criteria. 51 participants seeking neuraxial labor analgesia were randomly assigned to either the ID group (n = 25) or the SC group (n = 26) after providing informed consent. The Consolidated Standards of Reporting Trials (CONSORT) flow diagram for patients enrolled is shown in Fig 1 [20].

The median age of participants was 31.5 years, with 60.8 percent having a gravida of two or more, and median BMI of 30.6. The median NRS score for labor pain at the time of epidural analgesia request was seven. The two groups had no statistically significant differences regarding demographic and baseline characteristics (Table 1).

Pain responses to the lidocaine injection were assessed using CPOT and NRS, as summarized in Table 2. In the SC Group, the muscle tension component of CPOT was lower compared to the ID Group during lidocaine injection with a 25G needle by a statistically significant degree (1 [0,1] vs. 1 [1,1], p = 0.018). However, there were no other statistically significant differences between the groups in the overall median CPOT scores, the combined median scores for vocalization and body movement, or the median NRS scores. Furthermore, analyses using clinical pain cutoff scores (CPOT ≥ 2/8 or NRS ≥ 4/10) did not reveal any statistically significant differences between the groups.

Table 3 shows that the SC Group demonstrated a statistically higher NRS score during Tuohy needle insertion compared to the ID Group and the difference was statistically significant (3 [0,5] vs. 0 [0,3], p = 0.04). Despite this, the difference was not considered clinically significant since the median NRS scores for both groups remained below the "painful injection" threshold of four in both pain assessment scores. Similarly, patient satisfaction scores following epidural catheter placement did not differ significantly between the two groups (Table 3).

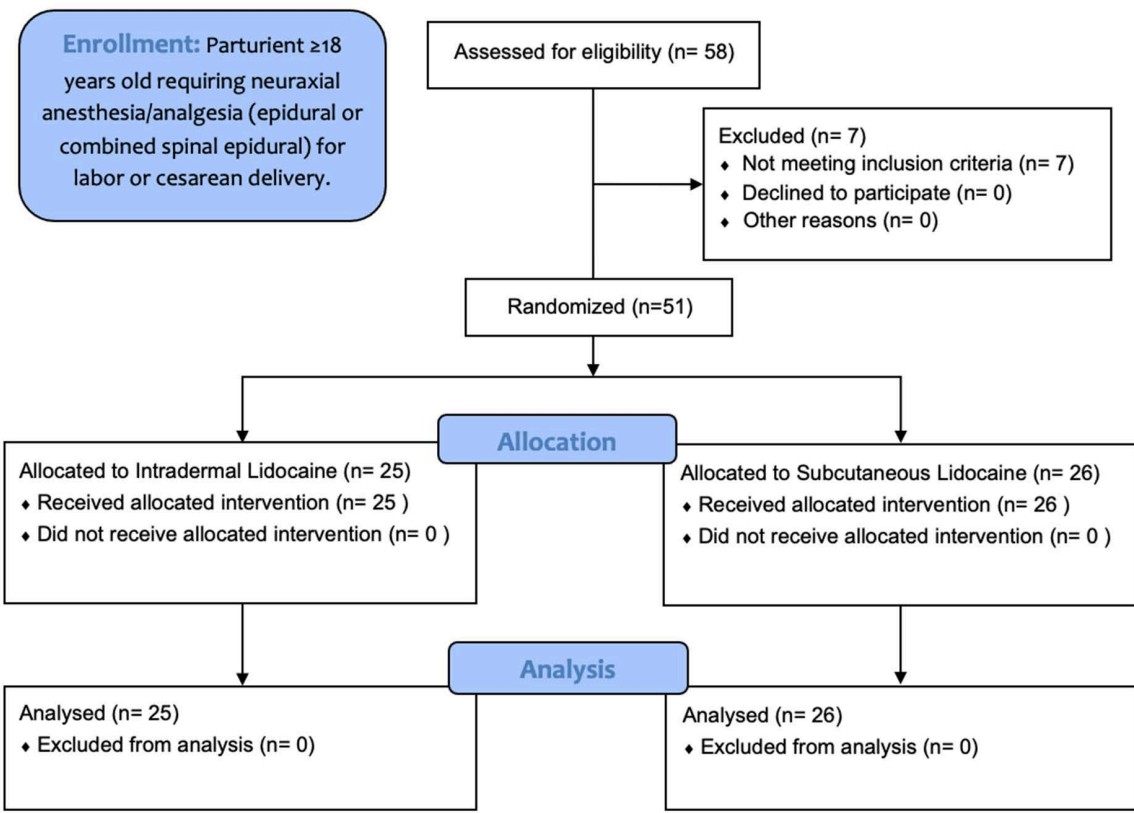

The above figure shows the randomization of 51 patients into two interventions: Intradermal Lidocaine (25 participants) and Subcutaneous Lidocaine (26 patients). All participants were analyzed according to original group allocation.

**Fig 1. CONSORT flow diagram of patient enrollment, allocation, and analysis.**

Hemodynamically, the mean HR, systolic BP, and diastolic BP in the SC Group after lidocaine injection and Tuohy needle insertion were lower when compared to ID Group, although these differences were not statistically significant (Table 4). Pairwise comparisons of hemodynamics were made between intradermal and subcutaneous injection techniques at three timepoints: baseline, after lidocaine injection, and after Tuohy insertion.

The Spearman correlation coefficient indicated a weak correlation between CPOT and NRS scores for the lidocaine injection ($r = 0.32$, $p = 0.024$) and the Tuohy needle insertion ($r = 0.47$, $p = 0.005$). Neither CPOT nor NRS scores were correlated with systolic BP changes before and after the lidocaine injection ($p > 0.05$).

No significant inter-provider variability was observed in patient satisfaction scores among the three unblinded anesthesia providers. Additionally, no significant inter-observer variability was found in the total CPOT scores assessed by the nine blinded examiners ($p > 0.05$).

## Discussion

This well-randomized pilot trial demonstrated comparable analgesic efficacy between intradermal and subcutaneous lidocaine for Tuohy needle insertion, with no significant differences in overall CPOT (primary outcome), or NRS scores, or

**Table 1. Demographics and characteristics of intradermal and subcutaneous groups.**

| | Intradermal (N = 25) | Subcutaneous (N = 26) | Total (N = 51) | P-value |
|---|---|---|---|---|
| Age (Mean (SD)) | 31.7 (6.7) | 31.2 (4.5) | 31.5 (5.6) | 0.53* |
| Ethnicity (N (%)) | | | | 0.362# |
| Not Hispanic or Latino | 21 (84.0%) | 24 (92.3%) | 45 (88.2%) | |
| Hispanic or Latino | 4 (16.0%) | 2 (7.7%) | 6 (11.8%) | |
| Race (N (%)) | | | | 0.942# |
| Asian | 3 (12.5%) | 4 (15.4%) | 7 (14.0%) | |
| Black or African American | 3 (12.5%) | 2 (7.7%) | 5 (10.0%) | |
| White | 15 (62.5%) | 17 (65.4%) | 32 (64.0%) | |
| More Than One Race | 3 (12.5%) | 3 (11.5%) | 6 (12.0%) | |
| Height in cm (Mean (SD)) | 163.1 (7.4) | 163.9 (5.8) | 163.5 (6.6) | 0.78* |
| Weight in kg (Mean (SD)) | 83.9 (18.1) | 80.4 (14.0) | 82.1 (16.1) | 0.54* |
| Body Mass Index (Mean (SD)) | 31.4 (5.6) | 29.8 (4.0) | 30.6 (4.9) | 0.23* |
| ASA (N (%)) | | | | 0.73# |
| II | 22 (88.0%) | 22 (84.6%) | 44 (86.3%) | |
| III | 3 (12.0%) | 4 (15.4%) | 7 (13.7%) | |
| Smoking (N (%)) | | | | 0.30# |
| Non-Smoker | 22 (88.0%) | 24 (92.3%) | 46 (90.2%) | |
| Smoker | 2 (8.0%) | 0 (0.0%) | 2 (3.9%) | |
| Former Smoker | 1 (4.0%) | 2 (7.7%) | 3 (5.9%) | |
| Gravida (N (%)) | | | | 0.49# |
| 1 | 11 (44.0%) | 9 (34.6%) | 20 (39.2%) | |
| ≥ 2 | 14 (56.0%) | 17 (65.4%) | 31 (60.8%) | |
| NRS Labor Pain Score when requesting labor epidural analgesia (Median (IQR)) | 8.0 (6.0, 8.0) | 7.0 (5.0, 8.0) | 7.0 (6.0, 8.0) | 0.26# |

\* Krushkal-Wallis p-value, # Chi-Square p-value, SD: standard deviation, IQR: interquartile range.

**Table 2. Pain evaluation of initial lidocaine injections via two techniques.**

| | Intradermal (N = 25) | Subcutaneous (N = 26) | P-Values |
|---|---|---|---|
| Primary Outcome: The Critical-Care Pain Observation Tool (CPOT) During Initial Lidocaine Injection (Median (IQR)) | | | |
| Total Scores | 3 (2, 4) | 2 (1, 4) | 0.18* |
| Vocalization & Body Movement | 1 (0, 1) | 1 (0, 1) | 0.68* |
| Vocalization | 0 (0, 1) | 1 (0, 1) | 0.31* |
| Facial Expression | 1 (1, 1) | 1 (0, 1) | 0.32* |
| Body Movement | 0 (0, 0) | 0 (0, 1) | 0.29* |
| Muscle Tension | 1 (1, 1) | 1 (0, 1) | 0.018* |
| Painful Injection, cutoff at ≥ 2/8 (n, (%)) | 21 (84.0) | 18 (69.2) | 0.21# |
| Numeric Pain Rating Scale (NRS) During Initial Lidocaine Injection (Median (IQR)) | | | |
| Total Scores | 3 (2, 4) | 3 (1, 5.5) | 0.98* |
| Painful Injection, cutoff at ≥ 4/10 (n, (%)) | 10 (40.0) | 11 (42.3) | 0.87# |

\* Krushkal-Wallis p-value, # Chi-Square p-value, SD: standard deviation, IQR: interquartile range.

**Table 3.** Analgesia efficacy via two techniques on tuohy needle insertions and patient satisfaction with epidural catheter placements.

| | Intradermal (N = 25) | Subcutaneous (N = 26) | P-Values |
|---|---|---|---|
| The Critical-Care Pain Observation Tool (CPOT) to Tuohy Needle Insertion (Median (IQR)) | | | |
| Total CPOT Scores | 2 (1, 2) | 2 (1, 3) | 0.53* |
| Vocalization & Body Movement | 0 (0, 1) | 0 (0, 1) | 0.71* |
| Vocalization | 0 (0, 1) | 0 (0, 1) | 0.84* |
| Facial Expression | 1 (0, 1) | 1 (0, 1) | 0.55* |
| Body Movement | 0 (0, 0) | 0 (0, 1) | 0.88* |
| Muscle Tension | 1 (0, 1) | 1 (0, 1) | 0.84* |
| Numeric Pain Rating Scale (NRS) to Tuohy Needle Insertion (Median (IQR)) | | | |
| NRS Scores | 0 (0, 3) | 3 (0, 5) | 0.04* |
| Painful Tuohy Needle Insertion (n, (%)) | | | |
| Total CPOT, cutoff at ≥ 2/8 | 13 (52.0) | 14 (53.8) | 0.89# |
| NRS, cutoff at ≥ 4/10 | 6 (24.0) | 12 (46.2) | 0.10# |
| Patient Satisfaction After Epidural Catheter Placement (Median (IQR)) | | | |
| Satisfaction Score | 10.0 (8.0, 10.0) | 10.0 (9.0, 10.0) | 0.12* |

\* Kruskal-Wallis p-value, # Chi-Square p-value, IQR: interquartile range.

**Table 4.** Hemodynamic changes of lidocaine injections via two techniques followed by tuohy needle insertion.

| | Heart Rates | | | Systolic Blood Pressure | | | Diastolic Blood Pressure | | |
|---|---|---|---|---|---|---|---|---|---|
| | Baseline | After Lidocaine Injection | After Tuohy Insertion | Baseline | After Lidocaine Injection | After Tuohy Insertion | Baseline | After Lidocaine Injection | After Tuohy Insertion |
| Intradermal (N = 25), Mean (SD) | 88.4 (15.5) | 91.6 (19.9) | 88.4 (18.7) | 125.2 (14.1) | 133.9 (19.4) | 133.0 (16.4) | 77.9 (12.9) | 81.2 (12.3) | 80.2 (10.6) |
| Subcutaneous (N = 26), Mean (SD) | 86.8 (11.92) | 85.8 (11.1) | 83.3 (11.9) | 133.3 (15.0) | 132.5 (14.9) | 128.3 (10.8) | 77.7 (10.2) | 78.2 (8.5) | 75.6 (8.3) |
| P-Values | 0.722* | 0.179* | 0.237* | 0.059* | 0.734* | 0.284* | 0.950* | 0.319* | 0.125* |

\* Linear Mixed Effect Model Pairwise Comparison P-Value, SD: standard deviation.

patient satisfaction. While CPOT and NRS showed weak but significant correlations for lidocaine injection and Tuohy needle insertion, neither correlated with systolic blood pressure changes. The subcutaneous technique significantly reduced CPOT-measured muscle tension and had lower hemodynamic responses post-procedure. Notably, no inter-provider variability in satisfaction or inter-observer variability in CPOT scores was observed, reinforcing methodological consistency and proof of principle for future research.

## CPOT vs. NRS: Observer-centric vs. patient-centric pain assessments

Our obstetric colleagues unintentionally use some components (e.g., muscle tension, body movements, vocalizations) of CPOT routinely to determine adequate anesthesia via painful stimulation with a hemostat before incision and during cesarean delivery under neuraxial blocks. To our knowledge, this is the first official application of CPOT in an obstetric population, extending its utility beyond critical care contexts [19]. Our study uniquely integrated the NRS, a subjective self-report tool, with the Critical-Care Pain Observation Tool (CPOT), an objective behavioral assessment, to evaluate pain during epidural procedures in laboring women. While NRS remains the gold standard for conscious pain reporting, CPOT,

validated here without significant inter-observer variability among nine blinded assessors, provides critical insights into nonverbal responses (e.g., muscle tension, body movements) that may influence procedural dynamics.

The weak-to-moderate correlation between CPOT and NRS scores (Spearman $r = 0.32$ for lidocaine injection; $r = 0.47$ for Tuohy needle insertion) highlights the distinct dimensions each tool captures: NRS reflects subjective pain perception, while CPOT quantifies observable physiological and psychological stress. Notably, this contrasts with prior ICU studies reporting stronger correlations ($r = 0.56$), likely due to differences in patient populations and procedural stressors [19,21]. Although our small sample size limited our ability to isolate the impact of specific CPOT components on provider performance, the tool's objectivity complements NRS by capturing labor-specific pain behaviors that patients may underreport.

These findings advocate the need for a hybrid assessment model in neuraxial procedures. While hemodynamic parameters and NRS alone may overlook nonverbal distress, CPOT enhances procedural safety by identifying clinically significant pain thresholds that could disrupt technique. Future studies with larger cohorts should explore how combined NRS-CPOT metrics can guide real-time procedural adjustments and improve patient-provider outcomes.

Despite the limited statistical power of this pilot trial, we observed a statistically lower muscle tension (a key component of the CPOT score) in the SC group compared to the ID group. This finding may be partially explained by the reduced pain sensitivity of subcutaneous injections, which are less affected by lidocaine injection rate and temperature due to histological differences in tissue structure suggested by a 1983 volunteer study [5].

Although Tuohy needle insertion pain was comparable between groups, extending the subcutaneous lidocaine injection's waiting period to 2–6 minutes could enhance its efficacy to match that of longer it takes to place intradermal injections [5,13]. Notably, patient satisfaction showed no significant variation across the three anesthesiologists performing epidural procedures, demonstrating study feasibility and justifying further investigation. Future research should prioritize a composite outcome score that integrates clinician-centered CPOT components, such as muscle tension (to avoid midline misidentification), body movements (to maintain critical positioning), and vocalizations (to minimize provider stress), alongside total CPOT and NRS scores.

## Limitations and challenges

This pilot study has several inherent limitations. First, the small sample size ($n = 51$) from a single center limits both the statistical power and the generalizability of the findings. This small sample size is a limitation for statistical significance. Second, the limited number of providers involved in the study makes it difficult to extrapolate the results to broader populations or other institutions. Third, the CPOT was originally designed for use in critical care settings and has not been validated for assessing pain during labor and delivery procedures; its reliability in this context remains to be established, particularly comparisons of individual components, such as muscle tension alone. However, this pilot study provides the basis for further validation. Fourth, parturients requesting epidural analgesia are often already experiencing significant labor pain, which may overlap with injection-related pain and complicate data collection. For instance, one patient in the SC Group of our study consistently rated injection/needle insertion as 10/10 on the NRS, regardless of variations in pain intensity. This may have confounded our statistical analysis and underscores the inherent challenges of subjective pain assessment in this setting.

## Conclusion

This pilot study demonstrates the importance and feasibility of using an objective pain assessment tool, CPOT, in future larger-scale research to further investigate acute pain during lidocaine administration. While overall pain scores (NRS, CPOT) and hemodynamic changes did not differ significantly between groups, subcutaneous injections resulted in statistically lower muscle tension, a CPOT component, compared to intradermal. Subcutaneous analgesia also matched intradermal in mitigating Tuohy needle insertion pain when assessed against "painful injection" thresholds, with comparable patient satisfaction. However, given small sample size, more research is necessary to validate these findings. This

study highlights the complementary roles of objective clinician-centric CPOT and subjective patient-reported NRS scores in evaluating procedural pain. CPOT may better reflect procedural impact and guide clinical decisions, warranting further validation in obstetric settings to optimize pain assessment beyond subjective tools alone. Additional research is required to gain deeper insights and to outline effective strategies for future implementation.

## Key Messages

Lidocaine is the most commonly used local anesthetic before Tuohy needle insertion for epidural catheter placement. But which administration method—subcutaneous or intradermal—causes less pain for laboring parturients? This study aims to answer that question using both an objective clinician-centric Critical-Care Pain Observation Tool (CPOT) and a subjective patient-centric Numeric Rating Scale (NRS). By comparing injection techniques, we seek to provide valuable insights into pain management strategies that could enhance patient comfort during epidural catheter placement.

## Supporting information

**S1 Data. Raw Data for RCT lidocaine administration techniques.** The complete raw dataset has been included as supporting information to ensure transparency and reproducibility. These files can be found in the supplementary materials section accompanying this work.
(XLSX)

**S1 Protocol. Initial submission approved protocol.**
(PDF)

## Acknowledgments

The authors gratefully acknowledge Jillian Tishko, BS, MPHc, Jeremy Reeves, BS, MSc, Chloe Evering, MDc, and Valeria Gamarra, MDc, for their regulatory and data collection support. Without them, it would not be possible to accomplish this project (they granted permission to be named in this publication).

## Author contributions

**Conceptualization:** Plato Lysandrou, Ling-Qun Hu.

**Data curation:** Lukas Croner, Haosheng Li, Andrew Devine, Yue Yu, Mahmoud Abdel-Rasoul, Elvia Vera Miquilena, Ling-Qun Hu.

**Formal analysis:** Yue Yu, Mahmoud Abdel-Rasoul.

**Funding acquisition:** Haosheng Li, Andrew Devine, Marco Echeverria Villalobos.

**Investigation:** Plato Lysandrou, Tyler Balon, Yun Xia, Ling-Qun Hu.

**Methodology:** Plato Lysandrou, Tyler Balon, Yun Xia, Nasir Hussain, Ling-Qun Hu.

**Project administration:** Alberto Uribe.

**Resources:** Marco Echeverria Villalobos, Alberto Uribe.

**Software:** Yue Yu, Mahmoud Abdel-Rasoul.

**Supervision:** Yun Xia, Mahmoud Abdel-Rasoul, Ling-Qun Hu.

**Validation:** Lukas Croner, Haosheng Li, Yue Yu, Elvia Vera Miquilena.

**Visualization:** Alberto Uribe.

**Writing – original draft:** Lukas Croner.

**Writing – review & editing:** Lukas Croner, Plato Lysandrou, Haosheng Li, Andrew Devine, Tyler Balon, Yun Xia, Nasir Hussain, Yue Yu, Mahmoud Abdel-Rasoul, Marco Echeverria Villalobos, Alberto Uribe, Elvia Vera Miquilena, Ling-Qun Hu.

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
