## [Decision Letter · Decision Letter 0]

3 Nov 2025

Dear Dr. Hu,

Thank you for submitting your manuscript to PLOS ONE. After careful consideration, we feel that it has merit but does not fully meet PLOS ONE’s publication criteria as it currently stands. Therefore, we invite you to submit a revised version of the manuscript that addresses the points raised during the review process.

We look forward to receiving your revised manuscript.

Kind regards,

Claudia Interlandi, Ph.D

Academic Editor

PLOS ONE

Journal Requirements:

2. In the online submission form, you indicated that the de-identified dataset generated and analyzed during the current study is available from the corresponding author upon reasonable request. The study protocol and statistical code used for analysis are also available upon request.

4. Please include a caption for figure 1.

6. We note that there is identifying data in the Supporting Information file <Initial Submission Approved. Pdf, Initial Submission Approved Protocol. pdf>. Due to the inclusion of these potentially identifying data, we have removed this file from your file inventory. Prior to sharing human research participant data, authors should consult with an ethics committee to ensure data are shared in accordance with participant consent and all applicable local laws.

-Location data

Additional Editor Comments :

The work is interesting, but further clarification and enrichment are needed before it can be considered for publication.

Reviewers' comments:

Reviewer's Responses to Questions

**Comments to the Author**

1. Is the manuscript technically sound, and do the data support the conclusions?

Reviewer #1: Partly

Reviewer #2: Yes

2. Has the statistical analysis been performed appropriately and rigorously?

Reviewer #1: Yes

Reviewer #2: Yes

3. Have the authors made all data underlying the findings in their manuscript fully available?

Reviewer #1: Yes

Reviewer #2: Yes

4. Is the manuscript presented in an intelligible fashion and written in standard English?

Reviewer #1: Yes

Reviewer #2: Yes

Reviewer #1: This was a pilot study to compare subcutaneous (SC) and intradermal (ID) lidocaine administration to evaluate lidocaine injection pain and its analgesic efficacy. As such, being a pilot, there was no formal statistical power analysis provided. It was basically proof of concept.

Clinical characteristics were compared between the study groups using a two-sample t-test or Kruskal- Wallis test for continuous variables and Chi-square test or Fisher’s exact test for categorical variables. For primary aims and perhaps appropriately given the small sample size, Kruskal-Wallis tests were used to test if there was a difference in CPOT score following lidocaine injection between the two study groups. For secondary aims, Kruskal-Wallis tests were used to assess the difference between the study groups in CPOT and NRS scores following Tuohy needle insertion, NRS score following lidocaine injection, as well as patient satisfaction scores. For hemodynamics, linear mixed-effects models were used to account for repeated measurements. Please comment on any repeated measures results, if any.

The results were for the most part obviously not statistically different. While overall pain scores (NRS, CPOT) and hemodynamic changes did not differ significantly between groups, subcutaneous injections resulted in statistically lower muscle tension. The limitations, one being sample size, were appropriately listed. Obviously, the investigators should caution the reader that the significant p-values, especially comparison of the muscle tension scores, should be interpreted with caution and should be mentioned in the limitations and challenges section.

Reviewer #2: This is an interesting study evaluating different ways to inject lidocaine during labor.

Major Comments

1. This is a very small study of 51 patients. The authors acknowledge this; however, I believe it is important to emphasize more clearly this major limitation.

2. The fact that such a small number of patients at a single center further underscores the major limitations of this study and the need for follow up study(ies) to confirm these preliminary results.

**Do you want your identity to be public for this peer review?** For information about this choice, including consent withdrawal, please see our Privacy Policy

Reviewer #1: No

Reviewer #2: No

---

## [Author Response · Author response to Decision Letter 1]

26 Nov 2025

Response to Reviewers

1. Comment: Please ensure that your manuscript meets PLOS ONE's style requirements, including those for file naming. The PLOS ONE style templates can be found at https://journals.plos.org/plosone/s/file?id=wjVg/PLOSOne_formatting_sample_main_body.pdf and https://journals.plos.org/plosone/s/file?id=ba62/PLOSOne_formatting_sample_title_authors_affiliations.pdf -

Response: We have revised the manuscript to comply with PLOS ONE’s formatting guidelines for both the main text and title/author sections.

2. Comment: In the online submission form, you indicated that the de-identified dataset generated and analyzed during the current study is available from the corresponding author upon reasonable request. The study protocol and statistical code used for analysis are also available upon request.

Response: The de-identified raw dataset has been uploaded as Supporting Information in accordance with PLOS ONE’s data-sharing policy. All personal identifiers have been removed to ensure participant privacy.

3. Comment: We note that the grant information you provided in the ‘Funding Information’ and ‘Financial Disclosure’ sections do not match.

Response: The funding details have been corrected and now match across the “Funding Information” and “Financial Disclosure” sections.

4. Comment: Please include a caption for figure 1.

Response: A descriptive caption for Figure 1 has been added and the figure re-uploaded: “The above figure shows the randomization of 51 patients into two interventions:

Intradermal Lidocaine (25 participants) and Subcutaneous Lidocaine (26 participants).

All participants were analyzed according to original group allocation.”

5. Comment: Please include captions for your Supporting Information files at the end of your manuscript, and update any in-text citations to match accordingly. Please see our Supporting Information guidelines for more information: http://journals.plos.org/plosone/s/supporting-information.

Response: Titles and captions for all Supporting Information files have been included at the end of the manuscript, and in-text citations have been updated accordingly.

6. Comment: We note that there is identifying data in the Supporting Information file <Initial Submission Approved. Pdf, Initial Submission Approved Protocol. pdf>. Due to the inclusion of these potentially identifying data, we have removed this file from your file inventory. Prior to sharing human research participant data, authors should consult with an ethics committee to ensure data are shared in accordance with participant consent and all applicable local laws.

-Location data

Response: All identifying information has been removed from the Supporting Information files. The uploaded dataset is fully anonymized and complies with participant consent and ethical standards.

7. Comment: If the reviewer comments include a recommendation to cite specific previously published works, please review and evaluate these publications to determine whether they are relevant and should be cited. There is no requirement to cite these works unless the editor has indicated otherwise.

Response: We have addressed all reviewer suggestions:

Clarified limitations, emphasizing the small sample size and single-center design.

Added statements in the limitations section conclusion to caution interpretation of significant findings and highlight the need for larger studies: “Third, the CPOT was originally designed for use in critical care settings and has not been validated for assessing pain during labor and delivery procedures; its reliability in this context remains to be established, particularly comparisons of individual components, such as muscle tension alone” and “However, given small sample size, more research is necessary to validate these findings.”

Clarified “repeat” measures for linear mixed effect models: “For hemodynamics, linear mixed-effects models were used to account for measurements at three time points per participant: baseline, after lidocaine injection, and after Tuohy needle insertion”

Added clarifying statement to results: “Pairwise comparisons of hemodynamics were made between intradermal and subcutaneous injection techniques at three timepoints: baseline, after lidocaine injection, and after Tuohy insertion.”

Also updated p-values in Table 4 for precision.

---

## [Decision Letter · Decision Letter 1]

10 Dec 2025

Rethinking Procedural Pain in Labor: A Comparison of Lidocaine Injection Techniques for Epidural Catheter Placement Assessed with an Objective Clinician-Centric Pain Score -A Double-Blind Randomized Controlled Trial

PONE-D-25-38198R1

Dear Dr. Ling-Qun Hu,

We’re pleased to inform you that your manuscript has been judged scientifically suitable for publication and will be formally accepted for publication once it meets all outstanding technical requirements.

Kind regards,

Claudia Interlandi, Ph.D

Academic Editor

PLOS One

Additional Editor Comments (optional):

Thank you for the improvements made to the manuscript. It will now be considered for publication.

Reviewers' comments:

Reviewer's Responses to Questions

**Comments to the Author**

Reviewer #1: All comments have been addressed

2. Is the manuscript technically sound, and do the data support the conclusions?

Reviewer #1: (No Response)

3. Has the statistical analysis been performed appropriately and rigorously?

Reviewer #1: (No Response)

4. Have the authors made all data underlying the findings in their manuscript fully available?

Reviewer #1: (No Response)

5. Is the manuscript presented in an intelligible fashion and written in standard English?

Reviewer #1: (No Response)

Reviewer #1: (No Response)

**Do you want your identity to be public for this peer review?** For information about this choice, including consent withdrawal, please see our Privacy Policy

Reviewer #1: No

---

## [Editor Report · Acceptance letter]

PONE-D-25-38198R1

PLOS One

Dear Dr. Hu,

I'm pleased to inform you that your manuscript has been deemed suitable for publication in PLOS One. Congratulations! Your manuscript is now being handed over to our production team.

Kind regards,

on behalf of

Professor Claudia Interlandi

Academic Editor

PLOS One